# A Comparative Study of the Synthesis and Hydrolysis of *sym*-Triaminobenzene Homologues

**DOI:** 10.3390/molecules27238595

**Published:** 2022-12-06

**Authors:** Irina A. Shchurova, Natalia A. Alekseyeva, Sergey V. Sysolyatin, Valeriy V. Malykhin

**Affiliations:** Laboratory for Chemistry of Nitrogen Compounds, Institute for Problems of Chemical and Energetic Technologies, Siberian Branch of the Russian Academy of Sciences (IPCET SB RAS), Biysk 659322, Russia

**Keywords:** trinitrobenzenes, catalytic hydrogenation, Pd/Sibunit, triaminobenzenes, hydrolysis, 2-methylphloroglucinol, 2,4-dimethylphloroglucinol, 2,4,6-trimethylphloroglucinol

## Abstract

Here, we investigated the synthetic processes for the methyl derivatives of sym-triaminobenzene and phloroglucinol, the essential chemical reactants coming into use in the production of dyes and pigments, and medicinal drugs for different purposes. The most eco-benign process for the synthesis of triamino derivatives involves the catalytic hydrogenation of corresponding nitroarenes. The present study investigated the hydrogenation of 2,4,6-trinitrotoluene, 2,4,6-trinitroxylene, and 2,4,6-trinitromesitylene over a Pd catalyst. A 1% Pd/Sibunit catalyst was found to be preferable to the 5% analogue with a preserved palladium loading because it shortens the reaction time and provides a higher yield of the target product. The hydrogenation in methanol (or mixed methanol/toluene) at 50–55 °C and 0.5 MPa pressure produced 2,4,6-triaminotoluene, 2,4,6-triaminoxylene, and 2,4,6-triaminomesitylene, which were isolated as sulfuric acid salts in 98, 91, and 97% yields, respectively. The hydrolysis process of the resultant salts was examined, and conditions leading to mono-, di-, and trimethyl derivatives of phloroglucinol (90, 77, and 82%, respectively,) were identified. The hydrogenation of the trinitrobenzene homologues in mixed 7:1 (*v*/*v*) acetone/water, followed by hydrolysis to the respective polyphenols, was explored. A successful result was achieved only for 2,4,6-trinitrotoluene. The catalyst activity was shown to decline negligibly throughout 10 cycles of reuse. 2-Methylphloroglucinol was synthesized in a high yield ranging from 85 to 91% calculated as 2,4,6-trinitrotoluene.

## 1. Introduction

Benzene trinitro derivatives are a key feedstock for the synthesis of the corresponding triaminobenzenes whose hydrolysis furnishes polyphenols. These derivatives, and others, are important synthetic products that are finding application in various sectors of the chemical and pharmaceutical industries. For instance, phloroglucinol (1,3,5-trihydroxybenzene) and 2-methylphloroglucinol (2,4,6-trihydroxytoluene) are used in the manufacture of dyes and pigments [1], various polymers and pharmaceutical substances of different purpose [2,3,4,5,6,7,8]. 

It is known that the nitro groups can be reduced by the chemical action of metals in HCl [8,9,10]. However, this method is characterized by a great amount of near non-disposable waste. Therefore, the focus of researchers is concentrated on catalytic hydrogenation using Pd-based catalysts such as Pd/C [11,12,13,14,15,16,17,18], Pd/Al_2_O_3_ [19], Pt/C [16], Rh/Pd-C [20], or NiRe [21,22,23]. Despite Raney nickel being cheaper, it features an enhanced pyrophorosity and requires a higher pressure and temperature for the process to be successful, whereas Pd catalysts can work under milder conditions, which is the merit of the Pd catalysts.

The synthetic processes for 1,3,5-triaminobenzene through the hydrogenation of 1,3,5-trinitrobenzene and 2,4,6-triaminobenzoic acid, as well as the hydrolysis of 1,3,5-trinitrobenzene to phloroglucinol, have been well-studied by several research teams and are reported in the recent review paper [24]. At the same time, there is relatively little information on the behavior of trinitrobenzene homologues, more specifically 2,4,6-trinitrotoluene, 2,4,6-trinitroxylene, and 2,4,6-trinitromesitylene, in the hydrogenation reaction. The hydrolysis process allowing the synthesis of the corresponding valuable methyl derivatives of phloroglucinol is also underexplored [8,13,25,26]. 

The present study performed a comparative evaluation of the hydrogenation processes of sym-trinitrobenzene homologues over a Sibunit-supported Pd catalyst and investigated the hydrolysis process of the resultant sulfuric acid salts of triaminobenzene derivatives.

## 2. Results and Discussion

The literature overview demonstrates that Pd/C (or Pd/Sibunit) containing 5% Pd is most frequently used in the hydrogenation of *sym*-trinitrobenzene and 2,4,6-trinitrotoluene. That said, fresh catalyst portions are added to the spent catalyst in the subsequent hydrogenation cycles to bring the catalyst back into the process [17,27,28,29]. However, we showed [18], by the example of the hydrogenation of *sym*-trinitrobenzene, that it is more efficient to use the catalyst with a lower Pd content while preserving its loading to the substrate. For instance, 1% Pd/Sibunit caused the 1,3,5-triaminobenzene yield to decline in the 5th cycle, while 5% Pd/Sibunit declined as early as the second cycle. Fresh catalyst additives were not used, and the process was carried out in methanol at 50–60 °C and a hydrogen pressure of 0.5–0.6 MPa.

This result is explained by the fact that the Sibunit, whose content in the depleted catalyst is higher, acts as a sorbent and protects the active metal centers from the negative effect of impurities inevitably resulting from the hydrogenation process.

Here, we also utilized a Pd/Sibunit catalyst containing 1% Pd (354 m^2^/g specific surfaces as per BET, 0.6 cm^3^/g adsorptive capacity of pores, 6.7 nm mean particle size). The catalyst was added in the amount of 50% to the substrate weight. The reaction was conducted in methanol at 50–55 °C and a pressure of 0.5 MPa. Since triaminobenzenes as the base are sensitive to oxygen, they were isolated as sulfuric acid salts by treating the methanolic hydrogenate with concentrated H_2_SO_4_ (2.1 mol/mol nitroarene) (Figure 1). Table 1 summarizes the comparative results for four trinitrobenzenes.

The data in Table 1 corroborate that the yield of the respective triaminobenzene is higher when the 1% Pd/Sibunit is employed under the same conditions. Whereas this difference is small for **1a** [18] (Table 1, Entries 1 and 2), it is more than 10% for **1d**; besides, the hydrogen absorption time shortens considerably (Table 1, Entries 10 and 11).

Because the solubility of **1c** and **1d** in methanol was significantly lower than that of **1a** and **1b**, the methanol/substrate ratio was raised to 10 and 15, respectively (Table 1, Entries 4 and 8). It was found for both substrates that a further increase in the methanol quantity led to a decline in the yields of salts **2c** and **2d**, which is likely due to a good solubility of the salts in the reaction mixture (Table 1, Entries 4–6, 8–10). When a 4:1 (*v*/*v*) methanol/toluene mixture was used, the yield of **2c** increased to 91% due to the lowered solubility (Table 1, Entry 7).

The comparison between the yields of **2a**–**d** suggests that the trinitrobenzene homologues are hydrogenated more selectively than the unsubstituted *sym*-trinitrobenzene upon a complete conversion of the substrate. Obviously, being electron donors, the methyls promote a decline in the partly positive charge of the carbon atoms associated with the amine group. This phenomenon diminishes the activity of the compound in nucleophilic substitution reactions leading to polycondensation [30].

The synthesized salts **2b**–**d** were subjected to hydrolysis at reflux (Figure 2). To prevent the salts from oxidation, the water was pre-deaerated by adding sodium sulfite. In each case, the hydrolysis time was picked up for each compound individually by tracing the kinetics via HPLC. Table 2 lists comparative data.

The control of the acidity of the medium showed that the pH of the reaction mixture in all the syntheses remained unchanged during the process and was 0. The complete substitution of all the amino groups in **2a** and **2d** was found to take place within the same time in high yields of 82–86% (Table 2, Entries 1, 7, and 8). Reaching the maximum yield of **3b** (90%) requires 24 h of reflux (Table 2, Entry 3).

The hydrolysis of **2c** was the slowest and had the lowest yield (76%) (Table 2, Entry 5). Assuming that the acidity of the medium is high for salt **2c**, we elevated the initial pH to 2 (Table 2, Entry 6). In this case, the pH grew to 5 by the end of the reaction, and the hydrolysis rate and the yield of **3c** concurrently decreased significantly (Table 2, Entries 4 and 6). The major reaction product was 2-amino-4,6-dihydroxyxylene **4**, which was isolated from the cooled reaction mixture by filtration (Figure 3). NMR spectroscopy (see Appendix A) showed that the amine group is located between the methyls of xylene. The substitution of this amino group is the limiting stage in the process.

The hydrolysis of **2c** was the slowest and had the lowest yield (76%) (Table 2, Entry 5). Assuming that the acidity of the medium is high for salt **2c**, we elevated the initial pH to 2 (Table 2, Entry 6). In this case, the pH grew to 5 by the end of the reaction, and the hydrolysis rate and the yield of **3c** concurrently decreased significantly (Table 2, Entries 4 and 6). The major reaction product was 2-amino-4,6-dihydroxyxylene **4**, which was isolated from the cooled reaction mixture by filtration (Figure 3). NMR spectroscopy (see Appendix A) showed that the amine group is located between the methyls of xylene. The substitution of this amino group is the limiting stage in the process.

We previously showed [18] that methanol is not an optimum medium for the hydrogenation of **1a**. Our comparative experiments in methanol, aqueous methanol, water, acetone, and acetone–water mixtures proved that the acetone–water mixtures in a volume ratio of 4:1 to 7:1 (*v*/*v*) exhibit a better solubilizing ability with respect to all of the reactants whereby the catalyst activity is considerably extended and product yield and quality are observed to be high throughout 20 cycles.

We decided to check the efficiency of that solvent system in the hydrogenation of trinitrobenzene homologues.

The process was run in the following manner. First, compounds **1b–d** were hydrolyzed using aqueous acetone as the solvent. Since upon reaction completion the water content of the solvent became even higher, it was unreasonable to isolate **2b–d** via the standard protocol by acidifying with H_2_SO_4_ and filtration because of the losses due to their solubility. Therefore, the hydrogenation product that had been separated from the catalyst was treated with a dilute H_2_SO_4_ solution, obtained by mixing calculated amounts of concentrated H_2_SO_4_ and water, and was immediately subjected to hydrolysis. By heating the resultant reaction mixture, acetone was evacuated using a concurrent condenser until the flask temperature reached 100 °C, and the concurrent condenser was then replaced by a reflux condenser to maintain the hydrolysis process for the required time (Table 2).

By the example of **2b**, we showed a relationship between the yield and quality of **3b** and the molar ratio of sulfuric acid/**1b**.

Figure 1 displays the yield of crude **3b** obtained by extracting **3b** from the hydrolyzate with ethyl acetate. The maximum yields of the crude product ranged from 1.8 to 2.2 mol H_2_SO_4_/mol **1b**. In this case, the HPLC data demonstrated a gradual decline in the major product content within a range of 1.6–2.6 mol H_2_SO_4_/mol **1b** due to the contamination of **3b** with byproducts. Given the product purity, the optimum ratio was 1.8 mol H_2_SO_4_/mol **1b**.

We tested the catalyst performance stability in the hydrogenation reaction of **1b** in aqueous acetone. For this, we performed a series of experiments with the reuse of the spent catalyst and compared the results with the previously obtained data on the hydrogenation of **1a** [18] (Table 3).

It was observed that the catalyst activity declined more intensively in the hydrogenation of **1b** than **1a** (Table 3, Entries **1**–**7**). However, this approach can considerably enhance the yield of **3b** compared to the previous results and is technologically simpler [17,31].

We performed hydrogenation with the subsequent hydrolysis of **1c** and **1d** under similar conditions. The data in Table 3 show that the yield of **3d** does not exceed 30**%**. At the same time, **3c** was not at all liberated in the pure form but was obtained as a resin (the yield measured by HPLC). Given the data in Table 2, wherein the high yields of 76.6 and 84**%** were achieved for those compounds by the hydrolysis of the crystalline salts, it is evident that the low yield was caused by the hydrogenation stage and by the low solubility of **1c** and **1d** or the intermediates in the water–acetone solution.

Theoretically, the hydrogenation of the keto group of acetone can also occur under reaction conditions. The resulting isopropanol can reduce the solubilities of the trinitro derivatives, intermediates, or impurities, causing the catalyst to deactivate. To this end, we analyzed the acetone removals from the hydrolysis stage of **2a**–**d** by gas chromatography and conducted an experiment in which the substrate was absent (the control). The comparison between the chromatograms of the model mixtures of acetone/isopropanol in ratios of 50/50 and 95/5 showed no formation of isopropanol in all cases, including the control solution.

## 3. Materials and Methods

### 3.1. Chemicals

*p*-Nitrotoluene (99%, Thermo Fisher Scientific, Basel, Switzerland), *m*-xylene (≥96%, “Reakhim”, Moscow, Russia), and mesitylene (99%, Thermo Fisher Scientific, Basel, Switzerland) were used without additional purification. Methanol (99.8%, J.T. Baker, Phillipsburg, NJ, USA) and toluene (99.5%, ECOS company, Staraya Kupavna City, Russia) were employed as solvents. Catalysts 1% Pd/Sibunit and 5% Pd/Sibunit were manufactured by the Center for New Chemical Technologies at the Boreskov Institute of Catalysis SB RAS (CNCT IC SB RAS, Omsk City, Russia). 2,4,6-Trinitrotoluene, 2,4,6-trinitroxylene, and 2,4,6-trinitromesitylene were synthesized by the nitration of *p*-nitrotoluene [32], *m*-xylene [33], and mesitylene [34], respectively.

### 3.2. Characterization Techniques 

CHNS analysis was carried out using a Thermo Fisher FlashEA 1112 (Thermo Fisher, Basel, Switzerland) elemental analyzer. Melting points were determined on a Stuart SMP 30 (Bibby Scientific Ltd., Staffordshire, UK) instrument. IR spectra were taken on an FT-801 FTIR spectrometer (Simex, Novosibirsk, Russia). NMR measurements were performed on a Bruker Avance III 500 (500 MHz for ^1^H, 125 MHz for ^13^C) or a Bruker AM-400 (400 MHz for ^1^H, 100 MHz for ^13^C) spectrometer (Bruker Corporation, Billerica, MA, USA). Chemical shifts δ in ppm were calibrated by using the residual nondeuterated solvents as the reference for the ^1^H and ^13^C NMR spectra.

### 3.3. High-Performance Liquid Chromatography

The purity of the isolated products was determined by HPLC using an Agilent 1200 instrument (Agilent Technologies, Santa Clara, CA, USA). Chromatographic conditions were: Hypersil ODS (5 μm) 100 mm × 2.1 mm I.D.; mobile phase A: water + 0.2% phosphoric acid, B: acetonitrile; gradient: 0–10 min from 2 to 50% B, 5 min 50% B; detection: UV 205 nm; flow rate: 0.25 mL/min; temperature: 25 °C; injection volume: 5 μL; and sample concentration: 0.2 mg/mL.

### 3.4. Gas Chromatography

Gas chromatography of the reaction products was performed on a Chromos GH-1000 chromatograph (OAO Sibur, Dzerzhinsk city, Russia) equipped with a flame ionization detector and a 25 m × 0.25 mm × 0.5 µm capillary column. Analysis conditions were: 190 °C evaporator temperature, 210 °C detector temperature; column temperature: from 30 °C (1 min) to 40 °C (2 min) at a 1 °C/min heating rate, and then from 90 °C (1 min) to 135 °C (1 min) at a 3 °C/min heating rate. Helium was used as the carrier gas (24 cm^3^/min), and the injection volume was 0.1 µL. The products were identified by the known retention times.

## 4. Experimental

### 4.1. General Procedure for Sulfuric Acid Salts of Triaminobenzene Derivatives **2b**–**d**


Into a steel autoclave fitted with a magnetic stirrer and a gas feed system was loaded **1** (7 g), methanol or mixed MeOH/toluene, and 1% Pd/Sibunit (3.5 g). The autoclave was pressurized and purged thrice with nitrogen gas and then with hydrogen gas under vigorous stirring, after which, the reaction mixture began to be heated to 50–55 °C. The autoclave hydrogen pressure was maintained at 0.5 MPa. The process completion was judged by the absence of hydrogen absorption. Upon the reaction completion, the whole mixture was cooled to 23–25 °C, the hydrogen gas was relieved, and the reaction mixture was purged thrice with nitrogen before the catalyst was filtered and washed on a filter with methanol (10 mL).

A methanolic solution of **2** was placed into a flask equipped with a stirrer and a thermometer and purged with nitrogen, and 95% H_2_SO_4_ (2.1 mol/mol **1**) was dispensed with vigorous stirring at no more than 15 °C. Precipitated **2** was collected by filtration and washed in situ with methanol (10 mL). The sediment was dried in vacuo at 60 °C for 2 h.

*Procedure A:* methanol (50 mL) was used.

*Procedure B:* mixed 4:1 (*v*/*v*) methanol/toluene (70 mL).

*Procedure C:* methanol (105 mL).


*2,4,6-Triaminotoluene disulfate (**2b**)*


The title compound was synthesized from **1b** by *procedure A*. Yield: 10.78 g (98.5%) as a beige amorphous powder. IR (KBr): 2953, 2874, 2603, 1562, 1506,1458, 1396, 1373, 1162, 1141, 1084, 1059, 973, 873, 797, 698, 612 cm^−1^. ^1^H NMR (500 MHz, DMSO-d_6_): δ 1.97 (s, 3 H, CH_3_), 6.47 (s, 2 H, Ar-H), 7.95 (bs, NH). ^13^C NMR (125 MHz, DMSO-d_6_): δ 11.4, 105.8, 113.1, 133.7. Calcd for C_7_H_11_N_3_·2H_2_SO_4_: C, 25.23%; H, 4.50%; N, 12.61%; S, 19.22%. Found: C, 25.31%; H, 4.60%; N, 12.52%; S, 19.43%.


*2,4,6-Triaminoxylene disulfate (**2c**)*


The title compound was synthesized from **1c** by *procedure B*. Yield: 9.18 g (91.1%) as a beige amorphous powder. IR (KBr): 2892, 2610, 1564, 1549, 1502, 1373, 1322, 1170, 1094, 1079, 1053, 875 cm^−1^. ^1^H NMR (500 MHz, DMSO-d_6_): δ 2.03 (s, 6 H, CH_3_), 6.61 (s, 1 H, Ar-H), 8.25 (bs, NH). ^13^C NMR (125 MHz, DMSO-d_6_): δ 12.5, 105.8, 114.2, 131.1, 146.1. Calcd for C_8_H_13_N_3_·2H_2_SO_4_: C, 27.67%; H, 4.90%; N, 12.10%; S, 18.44%. Found: C, 27.75%; H, 5.03%; N, 11.89%; S, 18.49%.


*2,4,6-Triaminomesitylene disulfate (**2d**)*


The title compound was synthesized from **1d** by *procedure C*. Yield: 9.63 g (97.2%) as a white amorphous powder. IR (KBr): 3075, 2903, 2676, 2595, 1610, 1567, 1508, 1471, 1396, 1190, 1073, 1023, 866 cm^−1^. ^1^H NMR (500 MHz, DMSO-d_6_): δ 2.13 (s, 9 H, CH_3_), 8.17 (bs, NH). ^13^C NMR (125 MHz, DMSO-d_6_): δ 13.1, 113.7, 134.4. Calcd for C_9_H_15_N_3_·2H_2_SO_4_: C, 29.92%; H, 5.26%; N, 11.63%; S, 17.73%. Found: C, 29.90%; H, 5.31%; N, 11.26%; S, 18.06%.

### 4.2. General Procedure for the Hydrolysis of the Sulfuric-Acid Salts of Triaminobenzene Derivatives **2b–d**


Into a round-bottom flask equipped with a reflux condenser, a magnetic stirrer, and a thermometer were placed distilled water (50 mL) and sodium sulfite (50 mg). Compound **2** (5 g) was added to the solution at 60 °C with stirring. The reaction mixture was refluxed for the required time. After the reaction was complete, the reaction mixture was filtered as hot and the filtrate was cooled to 20–25 °C. The filtrate was then saturated with sodium chloride, and the product was extracted with ethyl acetate or diethyl ether (4 × 15 mL) and washed with brine (5 mL). The extract was dried over Na_2_SO_4_ and the solvent was then removed in vacuo.

*Procedure A:* reaction mixture refluxed for 24 h and **3b** extracted with ethyl acetate.

*Procedure B:* reaction mixture refluxed for 35 h and **3c** extracted with diethyl ether.

*Procedure C:* reaction mixture refluxed for 14 h and **3d** extracted with ethyl acetate.


*2-Methyl phloroglucinol (**3b**)*


The title compound was synthesized from **2b** by *procedure A*. Yield: 1.9 g (90.3%) as a pale-brown crystalline powder, purity ≥ 96% as per HPLC; Mp 216–219 °C. IR (KBr): 3425, 3343, 2867, 2360, 1626, 1539, 1471, 1437, 1298, 1247, 1144, 1081, 1004, 945, 815, 800 cm^−1^. ^1^H NMR (400 MHz, DMSO-d_6_): δ 1.81 (s, 3 H, CH_3_), 5.77 (s, 2 H, Ar-H), 8.70 (s, 1 H, OH), 8.83 (s, 2 H, OH). ^13^C NMR (100 MHz, DMSO-d_6_): δ 8.5, 94.4, 101.1, 156.0, 156.9.


*2,4-Dimethyl phloroglucinol (**3c**)*


The title compound was synthesized from **2c** by *procedure B*. Yield: 1.7 g (76.6%) as a pale-brown crystalline powder, purity ≥96% as per HPLC; Mp 157–159 °C. IR (KBr): 3528, 3426, 2934,1633, 1612, 1513, 1461, 1432, 1383, 1293, 1248, 1201, 1153, 1086, 1018, 929, 859, 828, 812, 697 cm^−1^. ^1^H NMR (500 MHz, DMSO-d_6_): δ 1.88 (s, 6 H, CH_3_), 5.93 (s, 1 H, Ar-H), 7.72 (s, 1 H, OH), 8.58 (s, 2 H, OH). ^13^C NMR (125 MHz, DMSO-d_6_): δ 8.7, 94.6, 101.6, 153.2, 154.1.


*2,4,6-Trimethyl phloroglucinol (**3d**)*


The title compound was synthesized from **2d** by *procedure C*. Yield: 1.9 g (82.3%), as a beige crystalline powder, purity ≥ 96% as per HPLC; Mp 183–186 °C. IR (KBr): 3388, 2934, 1611, 1467, 1381, 1337, 1316, 1247, 1096, 1008, 865, 733, 673 cm^−1^. ^1^H NMR (500 MHz, DMSO-d_6_): δ 1.95 (s, 9 H, CH_3_), 7.59 (s, 3 H, OH). ^13^C NMR (125 MHz, DMSO-d_6_): δ 9.5, 103.9, 150.9.

### 4.3. Synthesis of 2-Amino-4,6-dihydroxyxylene (**4**) 

Into a round-bottom flask equipped with a reflux condenser, a magnetic stirrer, and a thermometer were placed distilled water (50 mL) and sodium sulfite (50 mg). Compound **2c** (5 g) was added to the solution at 60 °C with stirring. Then, the pH was brought to 2 with 25% aqueous ammonia, and heating was continued until boiling. The whole mixture was held for 12 h and then cooled to 4 °C. The precipitate was collected by filtration, washed with water, and air-dried. 

Yield: 1.3 g (57%) as a beige amorphous powder, purity ≥96% as per HPLC; Mp 182 °C (decomp.). IR (KBr): 3544, 3412, 3347, 2943, 2868, 2647, 1619, 1520, 1469, 1429, 1387, 1368, 1341, 1279, 1226, 1184, 1163, 1160, 797, 700, 594 cm^–1^. ^1^H NMR (400 MHz, DMSO-d_6_): δ 1.80 (s, 6 H, CH_3_), 4.25 (s, 2 H, NH_2_), 5.74 (s, 1 H, Ar-H), 8.40 (s, 2 H, OH). ^13^C NMR (125 MHz, DMSO-d_6_): δ 9.8, 94.5, 98.8, 145.9, 153.3.

### 4.4. General Synthetic Procedure for Polyphenols **3b**–**d** from Trinitrobenzenes **1b**–**d**


Into a steel autoclave fitted with a magnetic stirrer and a gas feed system was loaded **1** (7 g), acetone (44 mL), water (6 mL), and 1% Pd/Sibunit (3.5 g). The autoclave was pressurized and purged with nitrogen gas and then thrice with hydrogen gas under vigorous stirring, after which, the reaction mixture began to be heated to 50–55 °C. The autoclave hydrogen pressure was maintained at 0.5 MPa. The process completion was judged by the absence of hydrogen absorption. Once the reaction was complete, the whole mixture was cooled to 25–30 °C, the hydrogen gas was relieved, the reaction mixture was purged thrice with nitrogen, and the catalyst was filtered under nitrogen flow (10 mL). The filtrate was collected into a round-bottom flask and washed on a filter with mixed 7:1 (*v*/*v*) acetone/water (10 mL).

The flask containing a water–acetone solution of **2** was fitted with a magnetic stirrer and a concurrent condenser. The whole mixture was purged with nitrogen gas, after which the nitrogen gas feeding was disabled and a premix of concentrated H_2_SO_4_ in water (53 mL) (1.8 mol H_2_SO_4_/mol **1**) was poured into the flask. By doing so, the temperature of the reaction mixture rose to 40–45 °C. Then, nitrogen gas was fed in again and acetone was withdrawn with heating. Once the temperature in the mixture attained 100 °C, the nitrogen gas feeding was disabled and the concurrent condenser was replaced by a reflux condenser to continue the hydrolysis. The reaction mixture was refluxed for the required time. After the reaction was complete, the reaction mixture was filtered as hot and the filtrate was cooled to 20–25 °C. The filtrate was then saturated with sodium chloride, and the product was extracted with ethyl acetate or diethyl ether (4 × 15 mL) and washed with brine (5 mL). The extract was dried over Na_2_SO_4_ and the solvent was then removed in vacuo.

*Procedure A:* reaction mixture refluxed for 24 h and **3b** extracted with ethyl acetate.

*Procedure B:* reaction mixture refluxed for 35 h and **3c** extracted with diethyl ether.

*Procedure C:* reaction mixture refluxed for 14 h and **3d** extracted with ethyl acetate.


*2-Methyl phloroglucinol (**3b**)*


The title compound was synthesized from **1b** by *procedure A* Yield: 3.9 g (91.1%) as a pale-brown crystalline powder, with a purity ≥96%, as per HPLC.


*2,4-Dimethyl phloroglucinol (**3c**)*


The title compound was synthesized from **1c** by *procedure B* Yield: 2.89 g (64.7%) as a black resin, and 18.5% content (HPLC). 


*2,4,6-Trimethyl phloroglucinol (**3d**)*


The title compound was synthesized from **1d** by *procedure C* Yield: 1.2 g (27%), as a beige crystalline powder, with a purity ≥ 96%, as per HPLC.

## 5. Conclusions

Thus, the present study obtained and compared data from the hydrogenation of *sym*-trinitrobenzene homologues, such as 2,4,6-trinitrotoluene, 2,4,6-trinitroxylene, and 2,4,6-trinitromesitylene, in methanol over 1% Pd/Sibunit at 50–55 °C and 0.5 MPa pressure. The 1% Pd catalyst was shown to have superiority over the 5% analogue, as it reduces the reaction length, enhances the yield of triaminobenzenes, and retains the activity longer when recycled. Because of the methyl substituents, the trinitrobenzene homologues were discovered to be hydrogenated with a higher selectivity compared to unsubstituted *sym*-trinitrobenzene. Consequently, 2,4,6-triaminotoluene, 2,4,6-triaminoxylene, and 2,4,6-triaminomesitylene were synthesized as sulfuric acid salts in 98, 91, and 97% yields, respectively. The hydrolysis process of the resultant salts was examined and the optimum reaction time was identified. 2,4-Dimethyl phloroglucinol was found to have the lowest formation rate, and the limiting stage was the substitution of the amino group in 2-amino-4,6-dihydroxyxylene. 2-Methyl phloroglucinol, 2,4-dimethyl phloroglucinol, and 2,4,6-trimethyl phloroglucinol were synthesized in 89, 70, and 80% yields, respectively (calculated on nitroarene). The hydrogenation of the trinitrobenzene homologues in mixed 7:1 (*v*/*v*) acetone–water with subsequent hydrolysis to the respective polyphenols was also examined. The result was successful only for 2,4,6-trinitrotoluene. When reused, the catalyst activity was shown to decline negligibly throughout ten cycles. 2-Methyl phloroglucinol was obtained in a high yield of 85 to 91% calculated on 2,4,6-trinitrotoluene.

## Data Availability

Not applicable.

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
