# Peer review of "A Comparative Study of the Synthesis and Hydrolysis of *sym*-Triaminobenzene Homologues"

_molecules, 2022, doi:10.3390/molecules27238595_

Round 1
Reviewer 1 Report
The manuscript under review investigates the synthesis processes of methyl derivatives of phloroglucinol. This topic is highly relevant because polyphenols are of great interest as chemicals for the synthesis of bioactive substances, colorants and polymers. The interest in the said compounds is supported by the references cited by the authors. In the manuscript, the authors examined and described two successive stages of the preparation of polyphenols: catalytic hydrogenation of 1,3,5-trinitrobenzene methyl derivatives to the respective amino derivatives and hydrolysis thereof.
The authors also looked into the hydrogenation of three methyl derivatives of phloroglucinol in methanol by using a 1% Pd/Sibunit catalyst. The authors obtained excellent yields of the corresponding sulfuric-acid salts of triaminobenzenes for all the structures studied. The authors introduced an original and appealing approach that uses mixed acetone/water as the solvent because acetone can be competitive to trinitro arenes in the hydrogenation reaction. Nonetheless, the authors adduced evidence that the hydrogenation conditions were quite mild and hindered the progress of the competitive reaction. The proposed approach turned out to be successful for 2,4,6-trinitrotoluene and demonstrated that 2-methylphloroglucinol could be obtained in a good yield. That said, the authors proved the feasibility of multiple reuse of the spent catalyst, which makes the process more economical. The hydrolysis reaction of the resultant triaminobenzene sulfuric-acid salts was also practiced well by the authors.
The manuscript described an unusual, new product from the substitution of amino groups in 2,4,6-triaminoxylene. Moreover, the authors compared their findings with the analogous data for unsubstituted phloroglucinol, and discussed how the methyl groups in the aromatic ring influenced the features of the reactions in question within the scope of their research objective.
The conclusions made in the manuscript reflect the research results and are consistent with the set objective. Overall, the authors have done much work which is of scientific and practical importance.
Minor concerns to be addressed by the authors:
1. The number code of triaminobenzenes is not specified in Scheme 2. Given the text, it must be 3a-d? Please check and revise accordingly.
2. On page 4, the sentence: “Because no isolation of the salt occurred when the aqueous solvents were used, the hydrogenation product was acidified with sulfuric acid, diluted with water and subjected to hydrolysis after the catalyst was removed”. This sentence seems incorrect or inaccurate. What do the authors mean? Please rephrase, to make it clearer for the reader.
3. Subsection 4.4. “General synthetic procedure for polyphenols 3b–d from trinitrobenzenes 1b–d”. I noticed a mistake in the sentence: “The title compound was synthesized from 1db”. Should it be 1d? In the same subsection, the authors specified the hydrogenation temperature of 50-60 °С and the hydrolysis temperature of 100-102 °С, whereas Table 3 in the “Results and Discussion” section indicates the ranges of 50-55 °С and 102-103 °С, respectively. Please revise and be consistent throughout the text.
Author Response
The authors' response to Reviewer 1's comments has been uploaded as a PDF file.

Reviewer 2 Report
The paper entitled “A comparative study of synthesis and hydrolysis processes of 2 sym-triaminobenzene homologues” describes the synthesis of phloroglucinol derivatives through hydrolysis of amino derivatives that are obtained by catalytic hydrogenation of the nitro compounds.
The paper is clearly written with a presentation of the results in tables and graph.
The conclusions are in accordance with the results.
The materials and methods and the supplementary materials are adequate (only the supplier of the catalyst is missing and should be added).
In conclusion, this paper is ready for publication with the indication of the supplier of the catalyst.
Author Response
The authors' response to Reviewer 2's comments has been uploaded as a PDF file.
